# Process Steps for High Quality Si-Based Epitaxial Growth at Low Temperature via RPCVD

**DOI:** 10.3390/ma14133733

**Published:** 2021-07-03

**Authors:** Jongwan Jung, Baegmo Son, Byungmin Kam, Yong Sang Joh, Woonyoung Jeong, Seongjae Cho, Won-Jun Lee, Sangjoon Park

**Affiliations:** 1Hybrid Materials Center (HMC), Sejong University, Seoul 05006, Korea; 2Department of Nano and Advanced Materials Science, Sejong University, Seoul 05006, Korea; raehyang2@naver.com (W.J.); wjlee@sejong.ac.kr (W.-J.L.); 3Wonik IPS, Pyeongtaek 17709, Korea; bmson@wonik.com (B.S.); bmkam@wonik.com (B.K.); ysjoh@wonik.com (Y.S.J.); sangjoon.park@wonik.com (S.P.); 4Department of Electronics Engineering, The Graduate School of IT Convergence Engineering, Gachon University, Seongnam 13120, Korea; felixcho@gachon.ac.kr

**Keywords:** epitaxy, RPCVD, low temperature, Si, SiGe, impurity, in-situ cleaning, pre-cleaning, surface treatment

## Abstract

The key process steps for growing high-quality Si-based epitaxial films via reduced pressure chemical vapor deposition (RPCVD) are investigated herein. The quality of the epitaxial films is largely affected by the following steps in the epitaxy process: ex-situ cleaning, in-situ bake, and loading conditions such as the temperature and gaseous environment. With respect to ex-situ cleaning, dry cleaning is found to be more effective than wet cleaning in 1:200 dilute hydrofluoric acid (DHF), while wet cleaning in 1:30 DHF is the least effective. However, the best results of all are obtained via a combination of wet and dry cleaning. With respect to in-situ hydrogen bake in the presence of H_2_ gas, the level of impurities is gradually decreased as the temperature increases from 700 °C to a maximum of 850 °C, at which no peaks of O and F are observed. Further, the addition of a hydrogen chloride (HCl) bake step after the H_2_ bake results in effective in-situ bake even at temperatures as low as 700 °C. In addition, the effects of temperature and environment (vacuum or gas) at the time of loading the wafers into the process chamber are compared. Better quality epitaxial films are obtained when the samples are loaded into the process chamber at low temperature in a gaseous environment. These results indicate that the epitaxial conditions must be carefully tuned and controlled in order to achieve high-quality epitaxial growth.

## 1. Introduction

The epitaxial growth of silicon, silicon-germanium (SiGe) layers on a silicon substrate is a well-established technology for semiconductor fabrication, and has long been applied to the production of high-performance image sensors [1] and power devices [2]. Moreover, the addition of phosphorus or boron precursors along with the Si precursor enables the growth of Si:P or Si:B alloys with much higher P or B doping concentrations than can be obtained via ion implantation [3,4,5,6]. In addition to Si, Group 4 elements such as Ge, C, etc. can be applied to stress engineering [7,8,9,10]. This enables a source/drain (S/D) strain engineering in which SiC epitaxy is used to increase electron mobility by providing tensile strain, while SiGe epitaxy is used to increase hole mobility by providing compressive strain. Further, relaxed epitaxy technology can be used to grow Ge as a relaxed-Ge channel with very high hole mobility [11]. Currently, epitaxy technology is a key process for the continued scaling and increased functionality of integrated circuits. With respect to scaling, the currently used FIN field-effect transistor (FINFET) technology has been extended down to 5 nm, but the most suitable structure for the 3 to 2 nm scale downwards is probably the gate-all-around FET (GAAFET) [12,13]. A key process for implementing the GAAFET is the stacked epitaxial growth of Si and SiGe to construct multi-layers of nanosheets. 

The chemical vapor deposition (CVD) processes used for epitaxial growth includes ultra-high vacuum CVD (UHVCVD) [14], reduced pressure CVD (RPCVD) [15,16], and atmospheric-pressure CVD (APCVD) [17]. Among these, RPCVD has the advantage of combining a high throughput with high quality epitaxy. While several papers have reported the growth of crystalline Si or SiGe [18,19,20,21,22,23,24], there are a few reports on key process steps in low-temperature epitaxial growth via RPCVD. In addition, many reports using dichlorosilane (DCS) as Si source have been performed at high temperature above 700 °C [18,19,20,21,22,23] due to poor chlorine (Cl) desorption of DCS at low temperature. Even though high order silane precursors can be used at lower temperature [24,25,26,27], a cyclic deposition and etch routine are needed for selective epitaxial growth (SEG), making DCS the most common Si precursor for SEG of Si and SiGe. Epitaxial growth is performed via several sequential steps. The first step is pre-treatment of the wafers, which usually involves an initial ex-situ wet cleaning step, after which the wafers are loaded into a load lock chamber of the RPCVD system. From there, the wafers are moved via a transfer chamber into either an additional in-situ cleaning chamber or directly into the process chamber, where a final pre-cleaning step such as H_2_ bake takes place. Finally, the epitaxial growth occurs in the process chamber (Figure 1). Each of these process steps need to be well tuned to grow high-quality epitaxial films. In particular, the most representative interface impurities in Si and SiGe epitaxy are oxygen (O), carbon (C), and fluorine (F). The presence of O at the interface of Si and epitaxial layer leads to staking faults in the Si film [28], while C leads to the precipitation of silicon carbide and causes dislocation or staking faults in the Si lattice. Meanwhile, surface-adsorbed F is unstable, such that Si–F bonds are easily transformed into Si–H–O bonds during the extended queue time under ambient atmosphere, thus promoting re-oxidation after the surface cleaning [20,29]. 

Therefore, the use of RPCVD equipment and processes that are free from impurities, or at least suppress impurities as much as possible, are essential. In the present paper, several key process steps are examined using a 300 mm cluster RPCVD system at a temperature as low as 700 °C with DCS precursor. First, for the application of ex-situ cleaning, wet cleaning with dilute hydrofluoric acid (DHF), dry cleaning, and a combination of wet and dry cleaning are compared. Then the effects of in-situ bake with H_2_ at various temperatures, and the effects of two-step baking with H_2_ followed by HCl, are investigated. Finally, the effects of the temperature and environment of the process chamber at the time of loading the wafers are examined.

## 2. Experimental Details

The epitaxy equipment used in the present work is an industrial RPCVD tool with a planetary reactor for the simultaneous loading of up to five 300 mm wafers on each satellite mounted on a graphite susceptor (Wonik IPS Co., Pyeongtaek, Korea). The susceptor and satellites rotate in synchrony to provide epitaxially grown films with highly uniform thicknesses. The precursors are injected through the shower heads in the middle of the reactor and are discharged through the outlet at the edge of the susceptor. The detailed reactor structure and modeling of the resulting CVD growth have been provided in previous work [30,31]. Before loading the p-type (100) 300 mm wafers into the epitaxial equipment, the wafers were cleaned with DHF (single wafer spin wet etcher, Apollon, Zeus. Co., Hwaseong, Korea). Starting from a stock solution of 49% HF, two concentrations of DHF solution were prepared in deionized (DI) water, namely 1:30 DHF/DI water (1.58% DHF, 0.932 M) and 1:200 DHF/DI water (0.24% DHF, 0.143 M). 

These solutions were compared for the pre-cleaning of the Si wafer. The cleaning times were adjusted based on the thermal oxide etching of about 5 nm. After the spin etching, the wafers were spin rinsed with DI water and then spin dried.

In addition, the ex-situ 1:200 DHF wet cleaning process was compared with dry cleaning and with a combination of 1:200 DHF wet + dry cleaning. The ex-situ dry cleaning process was performed using NF_3_/NH_3_ plasma. The queue time in the atmosphere was minimized after dry cleaning, the samples were loaded into the chamber, and Si epitaxial films were grown under the same conditions as for the wet-cleaned wafer.

In another set of experiments, the effects of additional in-situ bake in the presence of H_2_ only, or H_2_ followed by HCl were compared. The (100) Si wafers were first subjected to ex-situ wet cleaning with 1:200 DHF or ex-situ dry cleaning. Unless otherwise noted, the cleaned wafers were moved from the transfer chamber to the process chamber under vacuum. The loaded wafers were then subjected to in-situ bake at 700 °C for 10 min or at 750, 800, or 850 °C for 5 or 10 min in the presence of H_2_ and/or HCl. After baking, the dichloro-silane (DCS, SiH_2_Cl_2_), germane (GeH_4_), H_2_, and HCl were used for Si, SiGe blanket and selective epitaxy at 700 °C under pressures of 10–100 Torr.

In further experiments, the effects of the temperature and environment of the process chamber at the time of loading the wafers were examined. In these experiments, all the wafers were subjected to ex-situ dry cleaning before loading into the equipment. After dry cleaning, the wafers were loaded into a load lock chamber and subjected to vacuum before being moved into a transfer chamber and, finally, into the process chamber. The temperature of the process chamber at the transferal stage (i.e., the loading temperature) was varied as 550, 600, 650, and 700 °C, and the conditions of the process chamber were either a vacuum or a gaseous atmosphere.

The thicknesses and quality of the resulting epitaxial films were examined via Cs-corrected scanning transmission electron microscopy (HRTEM; JEOL, JEM-ARM200F, Tokyo, Japan) and secondary ion mass spectroscopy (SIMS; IMS 7f, CAMECA, Gennevilliers, France). O (^16^O), F (^19^F), and C (^12^C) concentration profiles were measured with a SIMS Cameca IMS 7f microanalyzer by using a 6 keV Cs^+^ mass-filtered primary ions of an intensity of 12 nA. The beam was rastered over an area of 250 × 250 μm^2^ and the analysis area was 150 μm in diameter. The quality of the epitaxial SiGe film was measured by high resolution X-ray diffraction (HRXRD) using PANalytical X’pert PRO (Malvern, UK) with a 4 bounce symmetric Ge (220) monochromator and slits in front of the detector.

## 3. Results and Discussions

### 3.1. Ex-Situ Wet DHF Cleaning

The concentrations of impurities in the SiGe epitaxial films obtained after pre-cleaning with 1:30 DHF and 1:200 DHF are indicated by the SIMS results in Figure 2. The SiGe film grown after cleaning with 1:30 DHF exhibits an O peak with a concentration of about ~10^20^/cm^3^ at the interface between the SiGe film and the Si substrate, whereas the film grown after cleaning with 1:200 DHF exhibits a decreased concentration of ~10^19^/cm^3^ (Figure 2a). Similarly, the level of F impurity is less in the film grown after cleaning with 1:200 DHF (Figure 2b). This may be attributed to the formation of more Si–H bonds, and correspondingly fewer unstable Si–F bonds, on the Si surface during the longer etching time in the presence of the lower-concentration HF solution (1:200 HF). 

### 3.2. Comparison of Ex-Situ DHF Wet Cleaning, Dry Cleaning, and Combined Wet + Dry Cleaning

The SIMS profiles of the Si epitaxial films obtained after ex-situ wet pre-cleaning with 1:200 DHF, dry cleaning, and combined wet + dry cleaning are presented in Figure 3. Thus, in Figure 3a, the O profile of the epitaxial substrate that was wet cleaned with 1:200 DHF exhibits an impurity peak with a concentration of ~1.8 × 10^18^/cm^3^. By comparison, the intensity of the O peaks of the dry cleaned and wet + dry cleaned substrates are measurably decreased. Among the dry and wet + dry cleaning, the wet + dry have a lower O concentration at the interface. Meanwhile, the F profiles in Figure 3b reveal a major decrease in the level of F impurity for the dry and the combined wet + dry cleaning, compared to the wet-only treatment, while the C profiles in Figure 3c reveal no impurity peak for all three cleaning conditions. These results can be attributed to the fact that the Si surface becomes H-terminated and hydrophobic when pre-cleaning is performed using DHF, which can prevent pollution or oxidation in the ambient atmosphere. However, in the case of the wet-only cleaning, the amount of subsequent queueing time in the ambient atmosphere is inevitably increased by the need for rinsing with DI water followed by drying. This significantly increases the possibility of re-oxidation. Not only is the need for these processes eliminated in the case of dry cleaning, but also there is the advantage of being able to perform the in-situ dry cleaning in a vacuum chamber. Hence, dry cleaning is more advantageous than wet cleaning in terms of the queue time constraints. The TEM images of the interface between the Si epitaxial layer and the Si substrate of the epitaxial films grown after the three different pre-cleaning are presented in Figure 4. Here, the interface between the substrate and the epitaxial layer is easily distinguished in the sample that was subjected to wet cleaning only (Figure 4a), and becomes progressively less well-defined in the epitaxial films that were subjected to dry cleaning only (Figure 4b) and to the combined wet + dry cleaning process (Figure 4c). These results suggest that both the dry and combined wet + dry cleaning processes provide better cleaning efficiency than the wet cleaning only, and that the wet + dry cleaning is more effective than the dry-only cleaning.

### 3.3. In-Situ Hydrogen and HCl Bake Cleaning

As the final cleaning process before epitaxial growth, in-situ bake is very important for removing any residual surface oxide from the Si wafer. This residual oxide may result from incomplete ex-situ cleaning, or by re-oxidation during the numerous processing steps after ex-situ cleaning. Although the effectiveness of H_2_ bake increases with increased temperature, a higher bake temperature leads to undercutting or erosion of the SiO_2_ or Si pattern and to changes in the doping profile [19]. The SIMS O and F profiles of the Si epitaxial films obtained after wet pre-cleaning and subsequent H_2_ bake at various temperatures are presented in Figure 5a,b, respectively. The corresponding areal impurity doses are presented as a bar chart in Figure 5c. These data reveal that levels of O and F impurities at the interface gradually decrease as the temperature of H_2_ bake increases from 700 to 800 °C, and that these impurities peaks are completely removed at 850 °C. Figure 6 shows the TEM images of the epi-Si obtained after wet pre-cleaning and subsequent H_2_ bake at various temperatures. (a) 700 °C for 10 min, (b–d) 750, 800, 850 °C for 5 min. The interfaces between the substrate and the epitaxial layer are distinguished in 700, 750, 800 °C—samples as SIMS results. In Figure 7, the HRTEM images showing atomic lattices and FFT diffraction patterns for the 750 °C-sample show the crystalline structure of the Si-epitaxial layer. A multi-stack consisting of Si/SiGe/Si on Si substrate was also grown. Figure 8 shows the TEM images of Si cap (~10 nm)/SiGe (~14nm)/Si buffer (~5 nm) multi-stack layers grown on a Si substrate. That sample was grown after wet pre-cleaning (200:1 DHF) and subsequent H_2_ bake at 700 °C. HRXRD was measured to characterize the SiGe epi layer. It shows SiGe layer has a 32% Ge fraction and strained. 

The SIMS O and F profiles of the Si epitaxial films obtained after dry pre-cleaning and subsequent H_2_ bake at various temperatures are presented in Figure 9. In comparison to the above results for the wet DHF pre-cleaned samples (Figure 5), an overall decrease in the concentration of impurities is observed in the dry pre-cleaned samples (Figure 9), even when H_2_ bake is performed at 700 °C for 10 min. Finally, to further improve the effectiveness of low-temperature surface treatment, the H_2_ bake at 700 °C for 10 min was followed by an additional bake step in the well-known Si etchant gas HCl [19,32], and the results are presented in Figure 10. Here, a decrease in the level of O impurity is clearly seen compared to that obtained after H_2_ bake only. It is expected that the HCl bake removes any contaminated upper Si surface, thus effectively decreasing the level of impurity at the interface.

### 3.4. Process Chamber Loading Conditions

The SIMS O and F profiles of the epitaxial layers obtained under various loading temperatures are presented in Figure 11. Here, peaks in the O profile are hardly discernable at loading temperatures below 700 °C (Figure 11a). Moreover, the levels of F also decrease with decreased loading temperature (Figure 11b). This suggests that the surface Si–H bonds generated during the dry pre-cleaning process are readily broken under the high-temperature loading conditions. Therefore, low temperature loading is required for high quality epitaxial growth. 

Finally, the SIMS O, F, and C profiles of the Si epitaxial films obtained using different loading conditions (vacuum loading or gaseous loading), where all wafers were dry pre-cleaned, are presented in Figure 12. Here, the O impurity is seen to be suppressed under the gaseous loading environment compared to that observed under the vacuum loading environment. This is explained similarly to the effects of loading temperature (i.e., the gaseous environment in the process chamber at the time of loading the wafer is effective for preserving the Si–H bonding on the surface of the Si wafer). 

## 4. Conclusions

The present study examined the effects of three main epitaxy steps (ex-situ cleaning, in-situ bake and the wafer loading environment) upon the quality of Si-based epitaxial films obtained via reduced pressure chemical vapor deposition (RPCVD). With respect to ex-situ cleaning, dry cleaning gave better results than either wet cleaning with a 1:200 solution of dilute hydrofluoric acid (DHF) in deionized water or wet cleaning with 1:30 DHF. However, the best cleaning effect of all was obtained using a combination of wet (1:200 DHF) and dry cleaning. A multi-stack consisting of Si/SiGe/Si on a Si substrate grown after wet pre-cleaning (200:1 DHF) and H_2_ bake at 700 °C showed strained-pseudomorphic layers by HRXRD. With respect to the in-situ H_2_ bake, the levels of oxygen and fluorine impurities were found to decrease as the temperature increased from 700 to 800 °C, and no impurities peaks were observed at a bake temperature of 850 °C. Moreover, an additional baking step in HCl after the H_2_ bake was effective even at a temperature as low as 700 °C. Finally, with respect to the conditions of the process chamber at the time of loading the wafers, loading under a gaseous atmosphere at lower temperature gave better epitaxial film quality.

These results provide a valuable guideline for high-quality, low-temperature epitaxial growth in a cluster RPCVD system. The recommended steps should be carefully applied in to avoid compromising the quality and throughput of the epitaxy equipment as each step is closely related to the epitaxy throughput.

## Figures and Tables

**Figure 1 materials-14-03733-f001:**
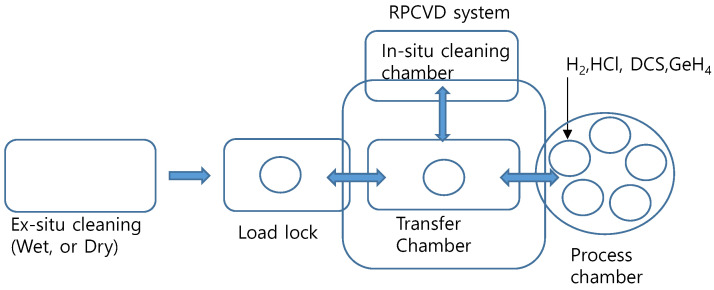
Basic schematic of epitaxial process flow using a reduced pressure chemical vapor deposition (RPCVD) tool.

**Figure 2 materials-14-03733-f002:**
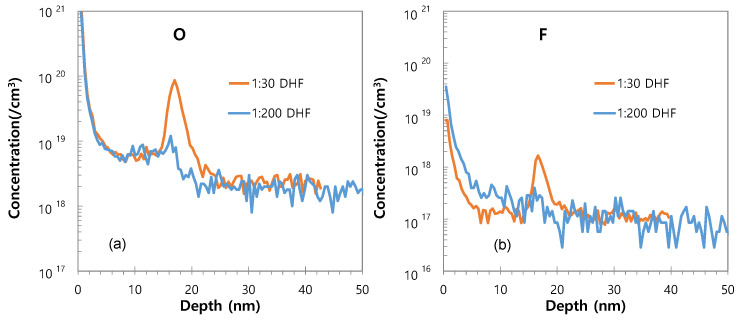
The secondary ion mass spectroscopy (SIMS) profiles of the SiGe epitaxial films obtained after pre-cleaning with 1:30 DHF and 1:200 DHF, indicating the concentrations of (**a**) O and (**b**) F.

**Figure 3 materials-14-03733-f003:**
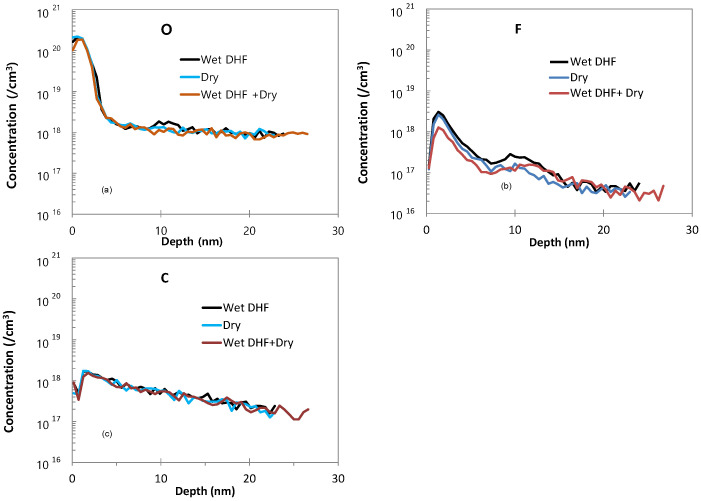
The SIMS profiles of the Si epitaxial films obtained after ex-situ wet pre-cleaning with 1:200 DHF, dry cleaning, and combined wet + dry cleaning, indicating the concentrations of (**a**) O, (**b**) F, and (**c**) C.

**Figure 4 materials-14-03733-f004:**
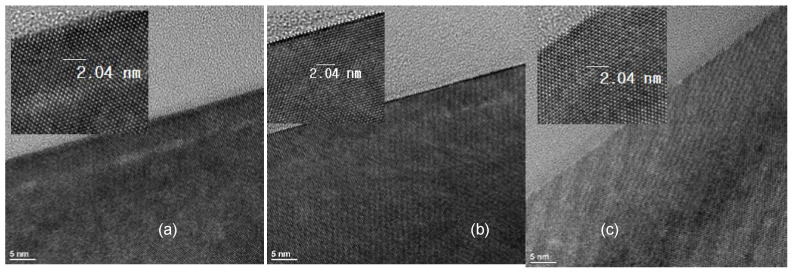
The TEM images of the Si epitaxial films obtained after ex-situ pre-cleaning with inset of HRTEM image (**a**) with 1:200 DHF (wet clean only), (**b**) dry clean only, and (**c**) wet + dry cleaning.

**Figure 5 materials-14-03733-f005:**
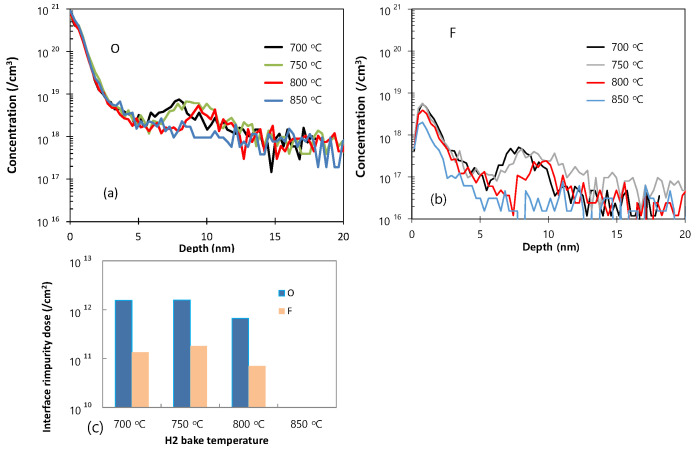
The SIMS O (**a**) and F (**b**) profiles of the Si epitaxial films obtained after wet pre-cleaning and subsequent H_2_ bake at various temperatures; (**c**) a bar chart showing the corresponding areal doses of O and F. (700 °C for 10 min, 750, 800, 850 °C for 5 min).

**Figure 6 materials-14-03733-f006:**
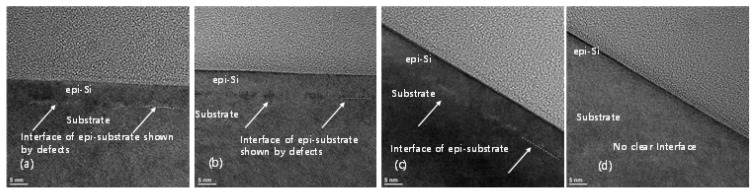
The TEM images of the Si epitaxial films obtained after wet pre-cleaning and subsequent H_2_ bake at various temperatures. (**a**) 700 °C for 10 min, (**b**–**d**) 750, 800, 850 °C for 5 min.

**Figure 7 materials-14-03733-f007:**
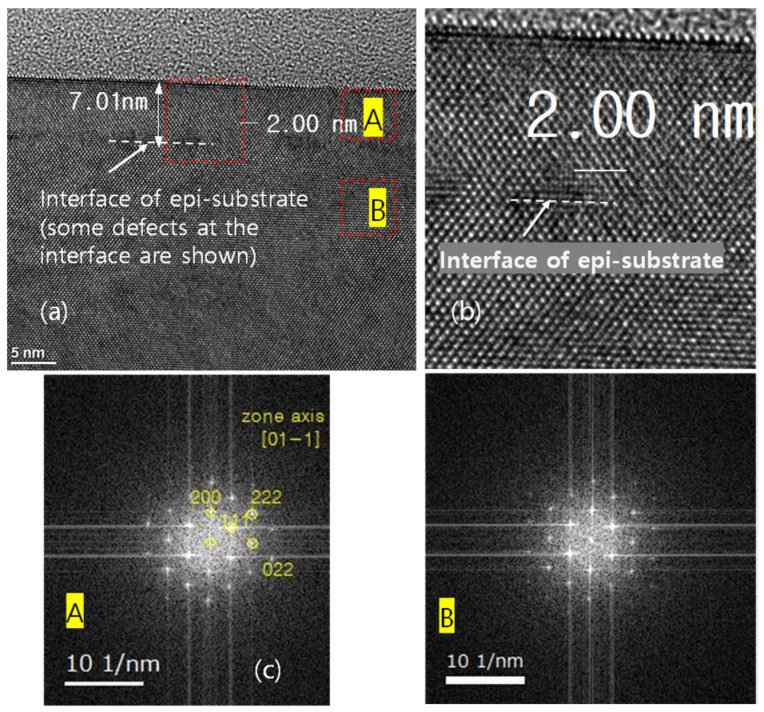
The TEM image of the epi-Si obtained after wet pre-cleaning and subsequent H_2_ bake at 750 °C 5 min. (**a**) TEM image, (**b**) TEM image of the red box, (**c**) FFTs of substrate and epitaxial layer showing crystalline structure.

**Figure 8 materials-14-03733-f008:**
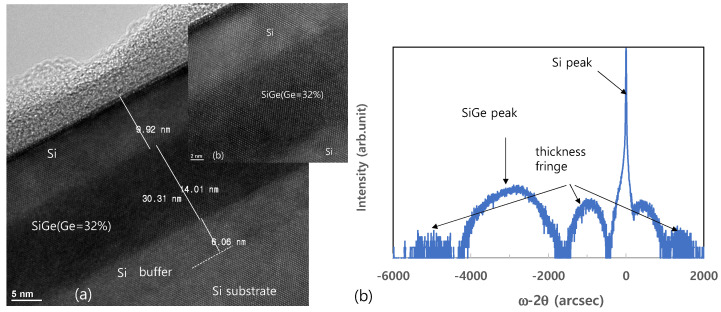
The TEM image and high resolution X-ray diffraction (HRXRD) of the Si (10 nm)/SiGe (14 nm)/Si (6 nm) epitaxial films obtained after wet pre-cleaning and subsequent H_2_ bake at 700 °C. (**a**) TEM image (**b**) HRXRD rocking curve (Si and SiGe peaks are shown).

**Figure 9 materials-14-03733-f009:**
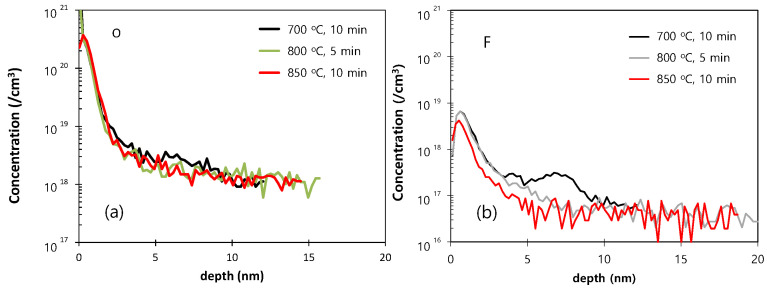
The SIMS O (**a**) and F (**b**) profiles of the Si epitaxial films obtained after dry pre-cleaning and subsequent H_2_ bake at various temperatures.

**Figure 10 materials-14-03733-f010:**
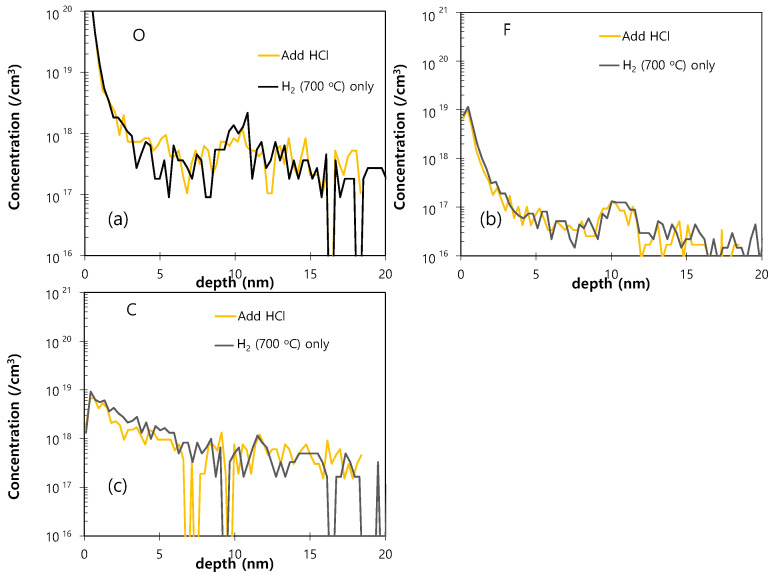
The SIMS O (**a**), F (**b**), and C (**c**) profiles of the Si epitaxial films obtained after ex-situ dry pre-cleaning and subsequent in-situ bake in the presence of H_2_ and then HCl at 700 °C.

**Figure 11 materials-14-03733-f011:**
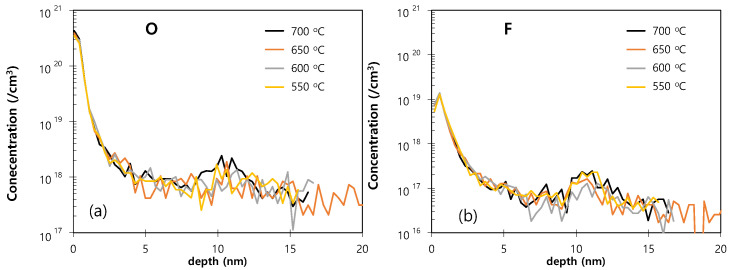
The SIMS O (**a**) and F (**b**) profiles of the Si epitaxial films obtained under various loading temperatures using samples that were dry pre-cleaned.

**Figure 12 materials-14-03733-f012:**
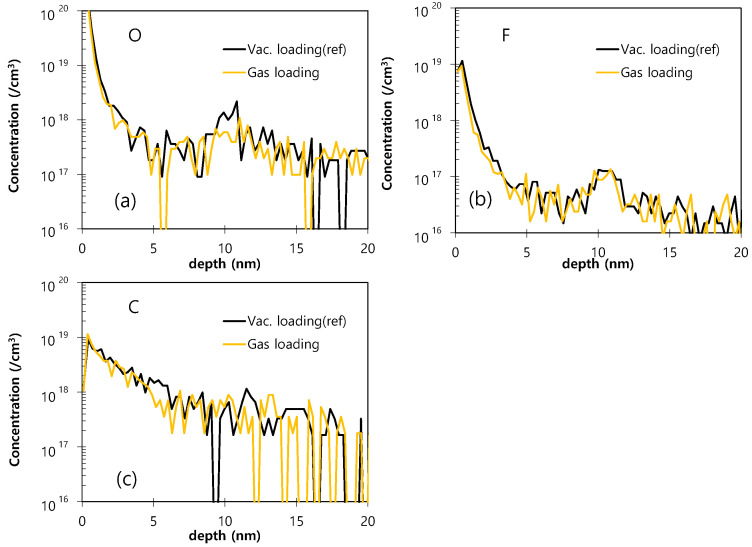
The SIMS O (**a**), F (**b**), and C (**c**) profiles of the Si epitaxial films obtained using different loading conditions (vacuum loading or gaseous loading), where all wafers were dry pre-cleaned.

## Data Availability

Data sharing is not applicable to this article.

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
