# Peer review of "Process Steps for High Quality Si-Based Epitaxial Growth at Low Temperature via RPCVD"

_materials, 2021, doi:10.3390/ma14133733_

Round 1

Reviewer 1 Report

The authors have described a method to grow Si-based epitaxial films obtained via reduced pressure chemical vapor deposition (RPCVD). It may be interested but can the authors explain why Figure 3 is a HR TEM, it is NOT high resolution as the atoms are not visible. It is not clear enough which substrate and which layer is. From this TEM images, the authors can not confirmed it is epitaxial oriented, Therefore I have doubt whether epitaxial Si film was formed. The authors need to introduce x-ray measurements to confirm this. If the authors can provide it is epitaxial oriented. It may be interested for the publication

Some minor comments:

Please introduce molar concentration in experimental section because it is unclear what the concentration is after diluting.

It was baking in the presence of H2 and/or HCl. But is this both in gas form or liquid? It is not clear to the readers. Baking is not the correct verb for the crystallization or growth step. Please use another verb

What was the working voltage of TEM?

Which specific signal of O and F were used in this work for SIMS? The description of SIMS device is lack

Reviewer 2 Report

This manuscript presents the results of the effect of the three main epitaxy steps (ex-situ purification, in-situ curing and wafer loading medium) on the quality of Si-based epitaxial films obtained by RPCVD. The study is useful. However, I still have quite a number of concerns in this manuscript, thus the manuscript can be reconsidered for publication after major revision.

1) I suggest adding a simple scheme when describing the steps of epitaxial growth via RPCVD for a better understanding of the information (the section “Introduction”).

2) There are now many studies of Si and SiGe epitaxial layers, including structural perfection. The authors should cite newer references when writing about impurities (section "Introduction").

3) Check the order of the references in the text. Reference [23] follows reference [27].

4) It would be very useful to compare the results with the state of the art of another similar study. If the research is original, the novelty of the it should be emphasized.

5) There should be much more discussion of the results in the manuscript, especially the authors need to provide sufficient physical mechanism in analyzing the results.

Author Response

Thank you.

Reviewer 3 Report

Work written correctly. The analysis of the test results is correct. However, I believe that additional research should be carried out for the publication to have greater scientific value. The work is sloppy and it is necessary to improve its graphic design. The drawing fonts, descriptions and scales should be the same. I appeal to the authors to carry out additional research and correct the drawings included in the work. 

Author Response

Thank you.

Round 2

Reviewer 1 Report

Thank you for the improving TEM images, and the authors have addressed on reviewers comments very careful. Two minor revisions:  Please index your XRD patterns on Figure 8B and add the scale bar on your FFT patterns, can the authors also calculate/identify the zone axis?

Reviewer 2 Report

The new version can be published

Reviewer 3 Report

I am asking you to correct only the paragraph in the conclusions. The authors enriched this publication. The paper is now suitable for the Materials science journal. 
